# Genetic reversal of the globin switch concurrently modulates both fetal and sickle hemoglobin and reduces red cell sickling

We previously reported initial clinical results of post-transcriptional gene silencing of BCL11A expression (NCT 03282656) reversing the fetal to adult hemoglobin switch. A goal of this approach is to increase fetal hemoglobin (HbF) expression while coordinately reducing sickle hemoglobin (HbS) expression. The resulting combinatorial effect should prove effective in inhibiting HbS polymerization at lower physiologic oxygen values thereby mitigating disease complications. Here we report results of exploratory single-cell analysis of patients in which BCL11A is targeted molecularly and compare results with cells of patients treated with hydroxyurea (HU), the current standard of care. We use single-cell assays to assess HbF, HbS, oxygen saturation, and hemoglobin polymer content in RBCs for nine gene therapy trial subjects (BCL$^{shmiR}$, median HbF% = 27.9) and compare them to 10 HU-treated subjects demonstrating high and comparable levels of HbF (HU High Responders, median HbF% = 27.0). All BCL11A patients achieved the primary endpoint for NCT 03282656, which was defined by an absolute neutrophil count greater than or equal to $0.5 \times 10^9$ cells/L for three consecutive days, achieved within 7 weeks following infusion. Flow cytometric assessment of single-RBC HbF and HbS shows fewer RBCs with high HbS% that would be most susceptible to sickling in BCL$^{shmiR}$ vs. HU High Responders: median 42% of RBCs with HbS%>70% in BCL$^{shmiR}$ vs. 61% in HU High Responders ($p = 0.004$). BCL$^{shmiR}$ subjects also demonstrate more RBCs resistant to HbS polymerization at lower physiologic oxygen tension: median 32% vs. 25% in HU High Responders ($p = 0.006$). Gene therapy-induced BCL11A down-regulation reverses the fetal-to-adult hemoglobin switch and induces RBCs with higher HbF%, lower HbS%, and greater resistance to deoxygenation-induced polymerization in clinical trial subjects compared with a cohort of highly responsive hydroxyurea-treated subjects.

Sickle cell disease (SCD) is an inherited hematologic disorder and an important genetic cause of childhood mortality in many countries, particularly in sub-Saharan Africa[1–3]. There are ~100,000 affected individuals in the United States, and despite improvements in SCD treatments including penicillin prophylaxis and hydroxyurea as standards of care, SCD is associated with significantly reduced life span due mainly to chronic and cumulative ischemia-induced end organ damage.

✉e-mail: david.williams2@childrens.harvard.edu; pablo.bartolucci@aphp.fr; higgins.john@mgh.harvard.edu

SCD is caused by a single amino-acid substitution in the β-globin gene, replacing a glutamic acid with valine, resulting in the expression of sickle hemoglobin (HbS : $\alpha_2\beta_2^S$) instead of the typical adult hemoglobin A (HbA : $\alpha_2\beta_2$). The hydrophobic valine allows HbS to polymerize under low physiologic oxygen tension. The presence of intracellular HbS polymer stiffens RBCs, damages their membrane and cytoskeleton, and precipitates a cascade of downstream pathology including vaso-occlusive, hemolytic, and inflammatory events. The clinical hallmarks include acute pain crises, chronic organ damage, decreased quality of life, and early mortality.

The only curative treatment for SCD currently is allogeneic hematopoietic stem cell transplantation (HSCT), ideally from an unaffected, HLA-identical sibling. For patients younger than 13 years, some studies indicate that this treatment has at least a 95% chance of cure, a 3-year survival >92%, and a 5-year probability of graft-vs-host disease (GVHD)-free survival of ~86–92%[4–6]. Unfortunately, donors are available for only a small fraction of patients, with one US study estimating only 18% of patients had a suitable donor[7]. HSCT with unrelated matched or mismatched donors is associated with significantly higher complications including acute and chronic GVHD and mortality[8].

The polymerization of HbS in a deoxygenated RBC is highly sensitive to the concentration of HbS and also depends on the fraction of fetal hemoglobin (HbF), the oxygen tension, and factors including 2,3-DPG concentration, temperature, and pH. The rate of HbS polymerization is extremely sensitive to small changes in HbS concentration[9,10]. The presence of HbF ($\alpha_2\gamma_2$) directly inhibits polymer formation more effectively than HbA[9–11]. Higher HbF% in RBCs inhibits HbS polymerization in vitro, and elevated HbF% in patients is associated with lower rates of morbidity and mortality[12–14]. Hydroxyurea (HU) increases HbF through an undefined mechanism and mitigates clinical symptoms to a varying degree for individual patients[15] and is the current standard of care for patients with severe SCD genotypes. Some patients continue to have significant vaso-occlusive events even with optimal HU dosage[15–20]. Gene therapy by gene addition to or gene editing of autologous hematopoietic stem cells (HSCs) has emerged as a promising curative treatment strategy[21–25].

Post-transcriptional silencing of BCL11A using lentivirus expression of a shRNA embedded in a microRNA architecture (shmiR) to re-activate γ-globin expression has been shown to be safe and to induce high levels of total HbF expression that is broadly distributed in RBCs in a pilot clinical study[22] leading to a multi-center phase 2 trial (NCT 05353647). Reversing the physiological fetal to adult globin switch may offer the advantage of simultaneously and coordinately increasing the expression of anti-sickling HbF and significantly reducing the expression of HbS in RBCs. This treatment approach may be distinguished from approaches that increase HbF expression pharmacologically or add sickling-resistant Hb transgenes without directly reducing HbS expression as well. The resulting combinatorial and coordinated reduction in HbS and elevation in HbF may further inhibit polymerization and increase the resistance of these RBCs to sickling at even lower physiologic oxygen tensions, such as in the renal medulla and bone marrow where oxygen tension may reach as low as 1% in healthy individuals[26,27].

Here we used state-of-the-art assays to measure HbF and HbS at the single-RBC level to determine whether RBCs derived from HSCs in which BCL11A was molecularly targeted showed a measurable decrease in HbS with increased HbF as compared to RBCs in patients with high HbF responses to HU who achieved comparable levels of total blood HbF%. To determine the biological relevance of this switch, we also compared the level of hemoglobin polymer in RBCs from BCL[shmiR] and HU subjects. Compared to HU patients, BCL[shmiR] patients had a higher percentage of RBCs rendered resistant to sickling at low physiologic oxygen tension.

## Results

### Participants and treatment

The pilot clinical trial aims to assess the safety and feasibility of gene therapy with the BCH-BB694 BCL11A shmiR-encoding lentiviral vector in patients with severe SCD. Details regarding the study protocol, eligibility criteria, and the CD34+ HSPCs collection, transduction, and infusion are provided in ref. 22, and consent for exploratory research reported here was embedded within the trial written informed consent (Boston Children's Hospital IRB protocol BCH #P00026188). The clinical trial (NCT 03282656) was performed under an approved Food and Drug Administration investigational new drug application (IND 17660) and was regularly reviewed by an independent data and safety monitoring board designated by the National Heart, Lung, and Blood Institute. Data reported were current as of April 2022. Participants did not receive compensation for their participation in this clinical trial.

This exploratory research report includes data from the first 9 patients treated in the pilot trial, averaging 32 months follow-up [range 17–49 months], including 8 patients with genotype HbSS and one with HbS/β[0]-thalassemia (Table 1). All 9 patients achieved the primary endpoint of the pilot clinical trial, which was the rescue of hematopoiesis after conditioning, defined by an absolute neutrophil count (ANC) $\geq 0.5 \times 10^9$ cells/L for three consecutive days, achieved within 7 weeks following infusion. Vector copy numbers (VCN) were determined by a quantitative polymerase-chain-reaction (qPCR) performed on erythroid precursors, B-lymphoid cells, and myeloid cells from peripheral blood and bone marrow at 6 months for each of the 9 patients (Table S2). VCN was stable after 6 months, and all 9 patients therefore achieved the pilot trial's secondary endpoint[28]. Aiming to investigate the cellular-level differences between the BCL[shmiR] and HU treatments for patients with similar levels of HbF, we focused analysis of the BCL[shmiR] patients on the 7 BCL[shmiR] patients (02, 04, 06, 07, 08, 09, 11) who did not receive RBC transfusions after infusion and engraftment of the BCH-BB694 BCL11A shmiR drug product (hereafter, Untransfused BCL[shmiR]). Transfused BCL[shmiR] refers to one patient who maintained periodic RBC transfusions due to Moyamoya present prior to gene therapy (patient 03) and one patient who received at least one intercurrent transfusion and had a lower overall HbF induction associated with a low in vivo VCN from the infused product (patient 10).

The hydroxyurea cohort included patients with sickle cell disease followed in one hematology clinic who were being treated with HU. Collection of samples from these patients was approved by the Boston Children's Hospital Institutional Review Board (BCH protocol #P00027349) and written informed consent was obtained from all patients who did not receive compensation for their participation in this study. Samples from 18 HU-responsive patients (median HbF = 24%, range 6–33) with more than 10 months of treatment (median time = 3.5 years, range 10 months–13 years) were obtained. We divided the HU cohort into two groups: patients with HbF higher than 20% (median HbF = 27%, range 23–33, $n = 10$, hereafter named HU High Responders), and the patients with HbF levels equal to or lower than 20% (median HbF = 12%, range 6–20, $n = 8$, hereafter named HU Low Responders). HU High Responders were selected solely on the basis of HbF% and not based on clinical outcomes, age, or alpha-thalassemia status, and thus the cohorts are not explicitly matched for any characteristics other than HbF%. This study was designed specifically to compare the molecular and cellular effects of treatments and not to compare clinical outcomes. Clinical outcome data is therefore not reported, and no claims are made about the superiority of these single-cell assays for predicting clinical outcomes compared to current standards. The threshold of 20% HbF was defined by the lowest HbF level observed for the 7 BCL[shmiR] patients (02, 04, 06, 07, 08, 09, 11), namely patient 02 had 20.8% of HbF at the 3.5-year post-gene therapy study visit. The basic hematologic characteristics of the three groups are compared in Supplementary Fig. S1 and are consistent with prior

**Table 1 | Hematologic data[a]**

| Treatment | Patient/cohort | Number of specimens/individuals n | Age interval Years | Time since infusion/HU treatment Months | Hb (g/dL) | MCH (pg) | MCHC (g/dL) | MCV (fL) | HbF (%) | F-cell (%) |
|---|---|---|---|---|---|---|---|---|---|---|
| Gene therapy | 2 | 7 | 21–25 | 49 | 10.9 (10.4,11.5) | 31.2 (30.9,31.8) | 35.0 (34.2,36.1) | 89.1 (88.3,91.6) | 23.3 (20.8,26.0) | 67.5 (59.7,76.5) |
| Gene therapy | 3 | 10 | 26–30 | 39 | 9.9 (9.1,11.1) | 30.0 (26.7,31.8) | 32.8 (32.3,35.7) | 89.2 (81.4,92.5) | 23.4 (16.5,30.1) | 62.6 (48.2,72.7) |
| Gene therapy | 4 | 9 | 21–25 | 40 | 11.1 (10.0,11.9) | 31.0 (30.2,32.2) | 36.4 (35.1,36.4) | 85.5 (84.2,88.3) | 28.9 (27.7,30.9) | 75.3 (70.2,81.9) |
| Gene therapy | 6[b] | 7 | 16–20 | 36 | 10.6 (10.5,11.1) | 25.8 (25.5,26.6) | 33.8 (32.2,34.1) | 77.8 (74.9,82.4) | 38.6 (36.1,40.1) | 71.8 (68.4,74.8) |
| Gene therapy | 7 | 8 | 11–15 | 32 | 11.8 (11.3,12.4) | 30.0 (30.0,30.1) | 36.1 (35.6,36.3) | 83.0 (82.8,84.4) | 25.5 (25.4,26.5) | 64.0 (63.1,67.0) |
| Gene therapy | 8 | 6 | 6–10 | 27 | 9.4 (9.0, 9.7) | 32.2 (31.3,32.7) | 36.5 (35.3,37.2) | 88.5 (86.9,95.8) | 37.8 (31.1,42.2) | 86.1 (80.8,91.1) |
| Gene therapy | 9 | 6 | 11–15 | 22 | 11.1 (9.6,11.4) | 38.4 (36.9,39.2) | 35.0 (34.8,35.5) | 109.8 (106.2,111.9) | 27.9 (25.3,28.9) | 75.4 (70.6,76.4) |
| Gene therapy | 10 | 6 | 16–20 | 23 | 7.6 (7.3,10.5) | 28.8 (28.2,30.3) | 35.0 (34.7,35.6) | 82.7 (81.5,85.2) | 11.1 (9.9,14.1) | 37.3 (30.0,45.4) |
| Gene therapy | 11 | 3 | 21–25 | 17 | 8.6 (8.4, 8.7) | 25.3 (25.1,26.0) | 35.3 (35.2,35.5) | 71.7 (70.8,74.0) | 27.6 (27.2,27.7) | 68.6 (64.9,69.8) |
| Gene therapy | Untransf. BCL^shmiR [c] | 9 | 16 (7,24) | 32 (17,49) | 10.9 (8.6,11.8) | 31.0 (25.3,38.4) | 35.3 (33.8,36.5) | 85.5 (71.7,109.8) | 27.9 (23.3,38.6) | 71.8 (64.0,86.1) |
| Hydroxyurea | HU High Responder | 10 | 7 (2,16) | 46 (10,156) | 10.2 (8.5,11.1) | 35.6 (25.8,40.7) | 36.0 (34.1,37.0) | 98.8 (75.5,110.4) | 27.0 (23.0,33.2) | 87.6 (78.1,93.6) |
| Hydroxyurea | HU Low Responder | 8 | 8.5 (3,23) | 66 (24,144) | 7.3 (6.1, 9.0) | 33.2 (26.0,36.8) | 35.0 (34.1,35.7) | 94.3 (75.2,104.2) | 11.7 (6.3,19.8) | 48.6 (29.9,75.2) |

[a] Values shown are median (minimum,maximum). Gene Therapy includes data ≥ 5 months after treatment. Subject ages are reported as 5-year intervals to maintain privacy. Source data are provided as a Source Data file.
[b] HbS/beta zero thalassemia.
[c] Untransf. BCL^shmiR corresponds to the seven untransfused BCL^shmiR patients: 2, 4, 6, 7, 8, 9, and 11.

reports for sickle cell patients[29]. See Table S1 for additional patient data.

Hb fractions in blood samples were determined by high-performance liquid chromatography (HPLC) or capillary electrophoresis (CE), the most popular methods for identifying and quantifying fractions of Hb variants[30, 31].

### Genetic targeting of BCL11A leads to a large induction of HbF determined in single RBCs

Untransfused BCL^shmiR subjects experienced an increase in overall fetal hemoglobin content to a median of 27.9% (Table 1) thus accounting for ~30% of the total cell hemoglobin content. HbF mass was quantified in individual RBCs using a recently developed flow cytometry protocol[32], and these measurements showed that the HbF response was driven by an increase in the fraction of RBCs with ≥10 pg HbF (Fig. 1a). HbF content in BCL^shmiR stabilized ~6 months after treatment. Among Untransfused BCL^shmiR subjects, a median of 50% of RBCs contained a minimum of 10 pg of HbF as compared to a median of 43% in HU-High Responders ($p = 0.1$, Fig. 1b). The 10 pg threshold has been postulated as a therapeutic goal[12] because it would reduce polymerization tendency in RBCs with MCHC ~ 33 g/dL when exposed to oxygen tension high enough such that >50% of the hemoglobin is saturated[12, 33, 34]. Higher levels of HbF would typically be required when MCHC is higher or oxygen tension is lower. Polymerization in a cell is exquisitely sensitive to the intracellular HbS concentration, making HbS concentration a better predictor of cell sickling than HbS mass[9,10]. RBCs with high HbF content will typically have lower levels of HbS and the HbS may be less likely to polymerize at a given oxygen concentration due to the anti-polymerization characteristics of HbF. We, therefore, compared the fractions of RBCs with HbS% above a specified threshold of 70% in the two cohorts. This fraction was inferred from the proportion of RBCs with HbF content lower than 30% of total MCH ignoring the minor hemoglobin A2 (HbA2) fraction. Figure 1c shows that the Untransfused BCL^shmiR subjects had fewer RBCs with high HbS% than subjects in the HU-treated cohort ($p < 0.004$). To define this cellular-scale difference more precisely, we next compared the Untransfused BCL^shmiR cohort to HU patients who had achieved blood HbF% > 20 (HU High Responders). Compared to this HU High Responders cohort with similar HbF content (median HbF = 27.0% for HU High Responders vs. median HbF = 27.9% for Untransfused BCL^shmiR, $p = 0.25$), the Untransfused BCL^shmiR patients had fewer RBCs with more than 70% HbS (Fig. 1c), consistent with the expected effects of reversal of the fetal to adult hemoglobin switch.

### Protection of RBCs from polymerization at low physiologic oxygen tension

We assessed hemoglobin polymer[35] in individual RBCs from BCL^shmiR subjects and HU-treated patients. RBCs in shear flow at physiological temperature and pH were exposed to oxygen tensions at physiological levels present in vivo in the bone marrow and kidney medulla (~1.7%) as well as in the peripheral microcirculation and in regions of the brain (~2.7%)[27]. We counted flowing RBCs with no detectable Hb polymer content (Fig. 2a) and those with detectable Hb polymer content (Fig. 2b). At these lower physiologic oxygen tensions, very high levels of HbF (>50%) are required to prevent Hb polymerization[9]. Figure 2c shows that Untransfused BCL^shmiR subjects had more RBCs with no detectable Hb polymer content at 1.7% oxygen tension compared to either the HU High Responders or HU Low Responders cohorts. The median Untransfused BCL^shmiR subject had 32% of RBCs with undetectable levels of polymer at 1.7% oxygen tension compared to 25% of RBCs with undetectable polymer for the median HU High Responders subject ($p = 0.01$), and 8% for the median HU Low Responders subject. A similar relationship was found at 2.7% (Fig. 2d), and no significant difference was observed at oxygen tensions equal to or higher than 3.7% (Fig. S2). Both groups have similar levels of overall blood HbF%

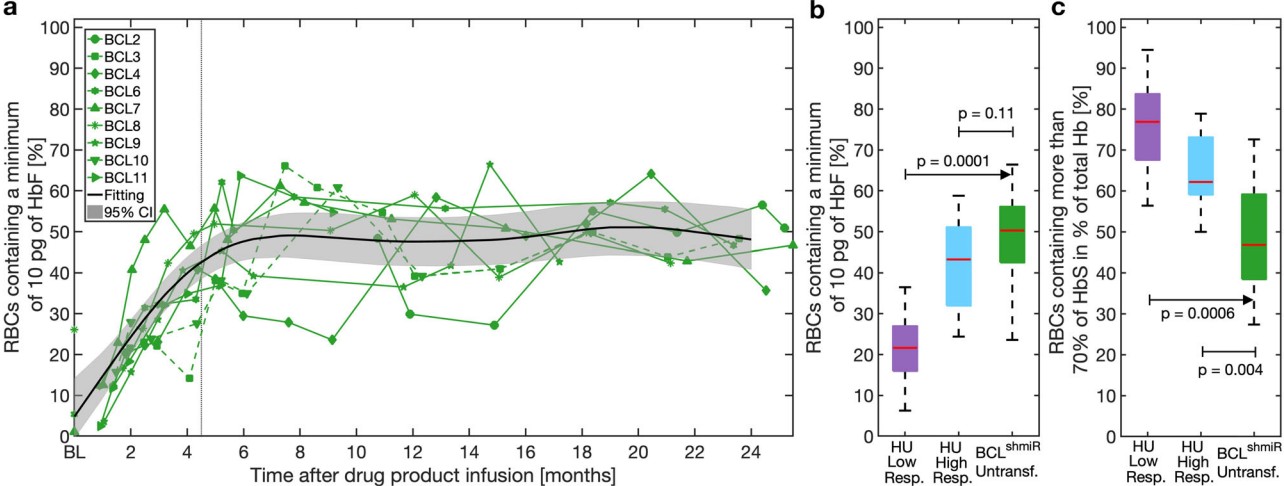

**Fig. 1 | HbF induction after BCL11A inhibition as compared to HU.** Panel **a** shows the proportion of RBCs containing a minimum of 10 pg of HbF per cell measured by flow cytometry in BCL^shmiR after drug product infusion. Dotted green lines represent Transfused BCL^shmiR (patients 3 and 10) while green solid lines represent the Untransfused BCL^shmiR patients at baseline (when available) and after drug product infusion. The vertical dashed gray line indicates 4.5 months after drug product infusion, which corresponds to timepoints for which previously transfused RBCs were no longer detectable (HbA = 0%) in Untransfused BCL^shmiR. For dotted green lines later than 4.5 months we show the percentage of RBCs containing a minimum of 10 pg of HbF among Untransfused RBCs. For timepoints earlier than ~4.5 months the percentage of RBCs containing a minimum of 10 pg of HbF reflects the presence of transfused RBCs. The black solid line shows a fitting for the Untransfused BCL^shmiR obtained by a linear mixed-effects model, and the gray region indicates the 95%

confidence interval for this linear mixed-effects model (see below). Panel **b** shows the difference in the proportion of RBCs containing a minimum of 10 pg of HbF between Untransfused BCL^shmiR (n = 7) in green, HU High Responders (n = 10), and HU Low Responders (n = 8) in purple (median–IQR). Panel **c** shows the proportion of RBC containing more than 70% of HbS as a percentage of total Hb, as determined by flow cytometry based on HbF quantification (excluding HbA2), in Untransfused BCL^shmiR after drug product infusion, for HU High Responders in blue, and HU Low Responders in purple. A linear mixed-effects model (described in the "Methods" subsection"Data analysis, statistics, and reproducibility") was used to estimate the mean kinetics of HbF induction as a function of time for the seven non-transfused subjects. Boxplot properties and the method used to compute the *p*-values in this study are described in the "Methods" subsection "Data analysis, statistics, and reproducibility". Source data are provided as a Source Data file.

---

(Fig. 2e), and these data are consistent with the observation that while HU High Responders patients have a larger overall percentage of F-cells (Fig. 2f), Untransfused BCL^shmiR subjects have a larger fraction of treatment-induced RBCs with the highest levels of HbF% per RBC than the HU High Responders subjects. Thus, this distribution of single-RBC HbF% in the Untransfused BCL^shmiR subjects appears to protect more RBCs from polymerization at lower physiologic oxygen tensions because fewer RBCs from HU High Responders subjects appear to contain the HbF levels needed to avoid polymerization at lower physiologic oxygen tensions.

### BCL targeting induces high HbF content per F-cell
We can estimate and compare the HbF content per F-cell by relying on the fact that most of the total blood HbF is contained in F-cells[36] and measuring HbF% and F-cell% with accepted clinical laboratory assays[37]. Figure 3a compares the ratio of total blood HbF% to F-cell% and finds that for Untransfused BCL^shmiR subjects the median HbF per F-cell was 40%, while the median for HU High Responders subjects was 32%. HbF per F-cell remained higher in BCL^shmiR than HU High Responders in analysis adjusted for total blood HbF% (Supplementary Fig. S3). This difference in HbF content per F-cell across this range of total blood HbF% is expected to yield a similarly systematic difference in the inhibition of intracellular hemoglobin polymerization particularly at lower levels of physiologic oxygen tension. Single-cell hemoglobin oxygen saturation and estimated polymer content were determined for all three cohorts at different oxygen tensions (Supplementary Fig. S4). For each oxygen tension, approximately one thousand RBCs were analyzed and classified based on whether oxygen saturation was preserved or reduced by polymerization which is more likely to occur at lower oxygen tensions. As expected from Fig. 2c and d, Untransfused BCL^shmiR showed fewer RBCs with significant amounts of polymer than HU High Responders (Supplementary Fig. S4b and d). Thus, despite fewer total F-cells

(Fig. 2f), BCL^shmiR treatment is associated with greater numbers of RBCs protected from polymerization at lower physiologic oxygen tensions. In addition, Fig. S3a and c show that the estimated degree of polymerization protection for the typical treatment-responsive RBC is greater for Untransfused BCL^shmiR compared to HU High Responders at 2.7% and 1.7% oxygen tensions. Thus, across the range of total blood HbF% studied, BCL^shmiR treatment appears to induce RBCs with higher HbF content per cell than HU High Responders, protect more RBCs from polymerization at lower physiologic oxygen tension, and provide a greater relative degree of protection from polymerization per treatment-responsive RBC (Supplementary Fig. S4).

### Hb switching after BCL11A inhibition concurrently reduces HbS cellular content compared with HbF induction after HU treatment
BCL11A inhibition is expected to reverse the fetal to adult hemoglobin switch, and this molecular change in treatment-responsive RBCs is expected to increase HbF expression and reduce HbS expression, leading to a greater anti-Hb polymerizing effect than would be expected from a treatment that only increases HbF expression. The green arrow in Fig. 3b depicts the Hb switch associated with a decrease in HbS/cell and an increase in HbF/cell. We performed a two-dimensional flow cytometric analysis to look for evidence of HbF-HbS switching in RBCs from the Untransfused BCL^shmiR patients compared with the HU High Responders cohort. For Untransfused BCL^shmiR subjects, pre-treatment (Fig. 4a) and post-treatment (Fig. 4b), and HU High Responders subjects (Fig. 4c) blood samples were analyzed. For HU High Responders subjects (Fig. 4c) the HbF/cell was shifted similarly to Fig. 4b, but there was no similar associated change in HbS/cell as observed for Untransfused BCL^shmiR subjects (Fig. 4b). Figure 4d shows that BCL11A inhibition treatment was associated with a significant relative reduction in HbS fluorescence intensity as HbF

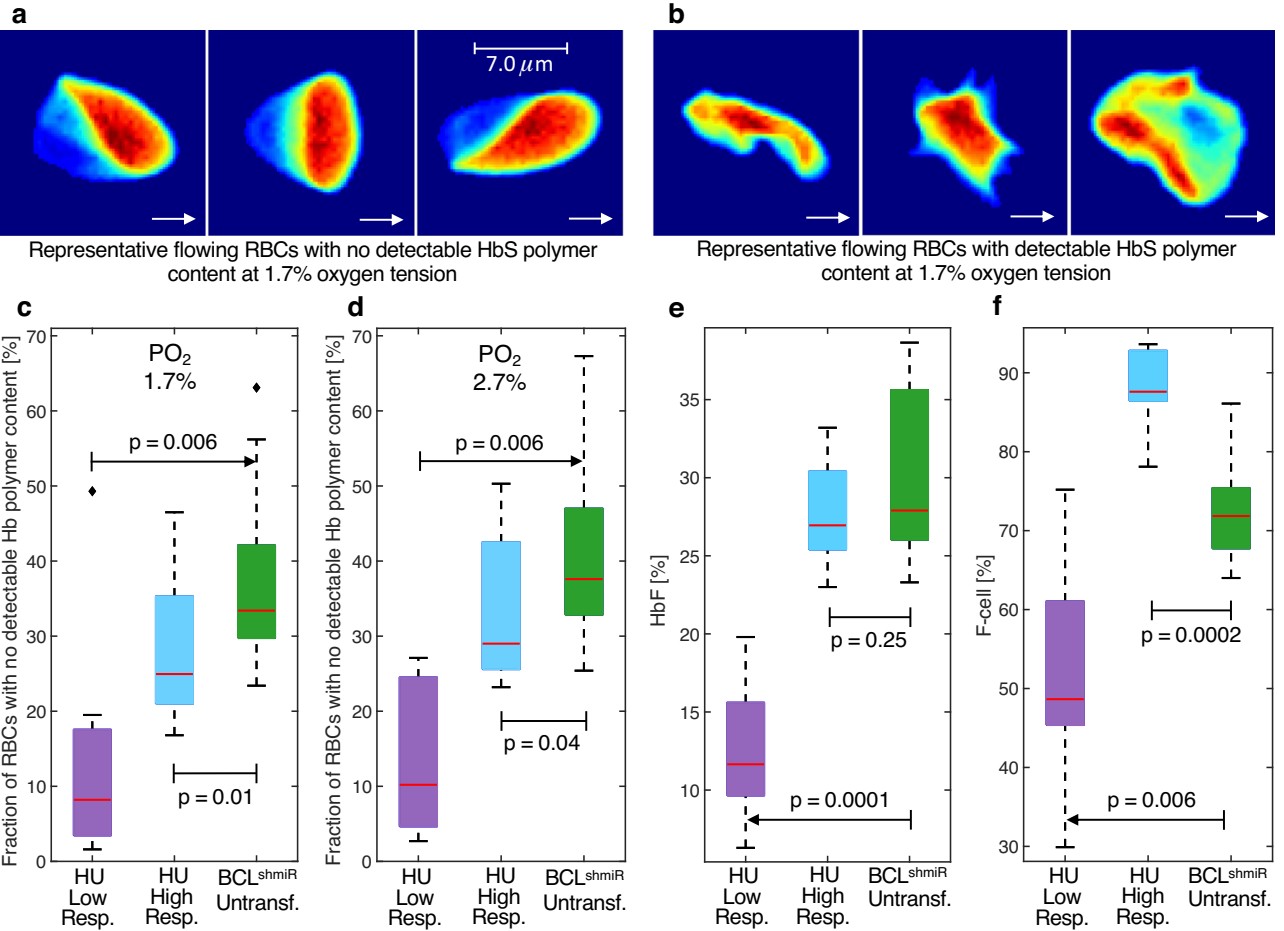

**Fig. 2 | Increased RBC fractions with no detectable Hb polymer content after BCL11A inhibition as compared to HU.** Panels **a** and **b** show representative RBC images from an HU-treated patient collected at 1.7% oxygen tension under shear forces in a microfluidic channel 20 microns wide and 8 microns deep. White arrows indicate the direction the cells were moving when the images were collected. Panel **a** shows RBCs with no detectable Hb polymer content, while panel **b** shows RBCs with a detectable polymer content. RBCs from panel **a** have most of their hemoglobin in the soluble state, and their shape is defined by their position relative to the wall (from left to right: closer to the top wall, at the center, and closer to the bottom wall) of the microfluidic channel and the resulting shear forces. RBCs from panel **b** have a significant amount of hemoglobin in the polymer state and their morphology (from left to right: elongated, holly leaf, and granular shapes) is altered by the dynamics of polymer aggregation upon deoxygenation. RBCs as shown in panels **a** and **b** were collected at 1.7% and 2.7% oxygen tensions for both Untransfused BCL$^{shmiR}$ and HU High Responders patients but with different fractions. Panels **c** and **d** show the fraction of RBCs with no detectable Hb polymer content at 1.7% and 2.7% oxygen tensions, respectively, measured in vitro in Untransfused BCL$^{shmiR}$ ($n = 7$; green boxes), in HU High Responders ($n = 10$; blue boxes), and in HU Low Responders ($n = 8$; purple boxes). Panels **e** and **f** show the HbF percentages (of total Hb) as measured by HPLC, and the proportion of F-cells as measured by flow cytometry. Boxplot properties and the method used to compute the *p*-values in this study are described in the "Methods" subsection "Data analysis, statistics, and reproducibility". Source data are provided as a Source Data file.

intensity increased. The HU High Responders cohort showed a significantly different pattern that was intermediate between the BCL11A inhibition pre-treatment and post-treatment patterns which is also reflected in the difference in mean cell hemoglobin content (MCH) in these groups (Table 1).

## Discussion

We report single-cell measurements of HbF, HbS, and oxygen-dependent Hb polymer and find that post-transcriptional silencing of BCL11A using a lentivirus vector expressing a shmiR reverses the fetal to adult hemoglobin switch and induces RBCs in patients that have increased HbF content and lowered HbS content. Untransfused BCL$^{shmiR}$ subjects have higher HbF content per F cell and more RBCs that are protected from polymerization and sickling at low physiologic oxygen tension than RBCs from a cohort of highly responsive hydroxyurea-treated patients who achieved similar levels of total HbF%. Across the range of total blood HbF% found in the Untransfused BCL$^{shmiR}$ cohort, we find evidence that HbF%-per-F-cell is higher

in Untransfused BCL$^{shmiR}$ subjects, and more RBCs are protected from polymerization at low physiologic oxygen tensions.

This study focuses on the cellular and molecular consequences of the reversal of the fetal to adult hemoglobin switch. While the early safety and clinical outcomes have been reported[22], the analysis reported here compared the cellular and molecular effects of treatment to those for HU in a cohort highly responsive to that treatment. The mechanism of action of HU has been hypothesized to involve differential effects on CFUe and BFUe which express significantly different levels of HbF[38,39] but is not fully defined, and further studies with RNA sequence analysis or other molecular data are needed to determine to what extent it acts via reversal of the fetal to adult switch. Since many patients with effective HbF induction on HU continue to experience vaso-occlusive events[15–20] and since the polymerization of deoxygenated Hb in sickle cells is highly dependent on HbS concentration, the combinatorial effect of increasing the HbF content while coordinately decreasing HbS content by targeting the physiologic switch is noteworthy and suggests the value of a direct

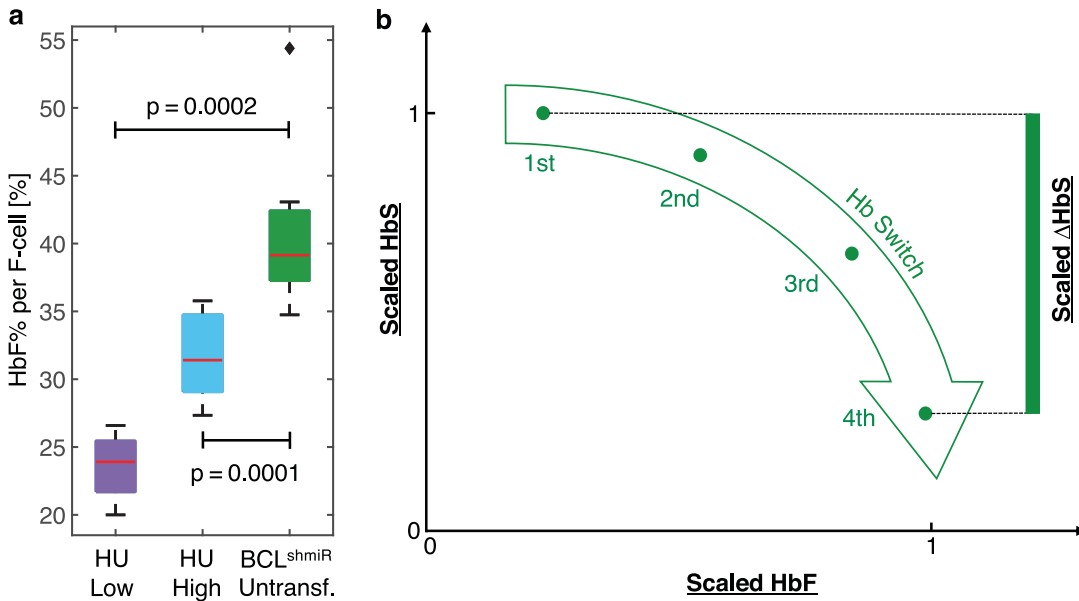

**Fig. 3 | BCL11A inhibition induces higher levels of HbF content per F-cell compared to HU and suggests an effective reversal of the fetal-to-adult Hb switch.** Box plots in panel **a** show HbF expressed as the percentage of total HbF per F-cell for Untransfused BCL$^{shmiR}$ ($n = 7$; green), HU High Responders ($n = 10$; blue), and HU Low Responders ($n = 8$; purple). The percentages of total HbF per F-cell are estimated as the ratio of total HbF% to F-cell% and assume that the HbF content in the non-F-cells is negligible. Panel **b** shows a schematic of the expected result of affecting the physiological Hb switch, with the green arrow representing the Hb switch path where HbF content increases while HbS content decreases. The $y$-axis and $x$-axis correspond to normalized HbF and HbS levels, for instance from scaled flow cytometric fluorescence intensities as shown in Fig. 4 with the green dots representing quartiles of the single-cell fluorescence intensity distribution. Boxplot properties and the method used to compute the $p$-values in this study are described in the "Methods" subsection "Data analysis, statistics, and reproducibility". Source data are provided as a Source Data file.

comparison of the clinical courses of patients treated with gene therapy targeting BCL11A and patients treated with HU who are optimally responsive to HU with adjustment for other potential confounders including patient age, gender, and genotype.

## Methods

### Study design and ethical oversight
This study was conducted in accordance with the principles of the Declaration of Helsinki (2013), and all relevant information and study documentation are provided in the section "Participants and treatment".

### Determining HbF mass thresholds of single RBCs
Single-RBC HbF content was measured by flow cytometry using a previously published protocol[32]. The protocol involves calibrating the fluorescence intensity of an anti-HbF antibody (against γ-globin−Reagent F−monoclonal antibody to human fetal hemoglobin conjugated with R-PE−from the Fetal Cell Count™ kit−IQ Products−IQP-363) and quantifying its correlation and the mean single-cell HbF content using RBCs from individuals with homogeneous HbF distributions. The anti-HbF antibody was used as per the manufacturer's instructions, using 50 μL of the antibody diluted with 50 μL of fixed and permeabilized cell suspension and incubated for 15 min, shielded from light at room temperature. The protocol constructs a standard curve for each batch of anti-HbF antibodies and a normalization of the fluorescence intensity by beads. Each standard curve was required to have a significant correlation with $R^2 \geq 0.95$. Intracellular co-staining using anti-HbF and anti-HbS (against β$^S$-globin−anti-HbS mouse monoclonal antibody−Rockland−200-301-GS5) was performed to assess the relative levels of HbS and HbF in single RBCs. For co-staining, anti-HbF was used diluted as above and anti-HbS was used at a final concentration of 20 μg/mL, incubated for 15 min, and shielded from light at room temperature. Control experiments were performed to ensure that the anti-HbS antibody and any HbS polymer did not

interfere with the HbF measurement (Supplementary Fig. S5a). This study uses this protocol to compare the relative levels of HbS and HbF staining intensity in order to assess qualitative evidence in favor of globin switch reversal and does not estimate single-RBC HbS mass. Control experiments also showed that the presence of polymer does not significantly interfere with the HbF measurement (Supplementary Fig. S5b). The gating strategy applied to exclude doublets and to analyze HbS+ RBCs in case of transfusion is described in Supplementary Fig. S6. Flow cytometry data was acquired using BD FACSDiva Software 9.0.1 (Beckton Dickinson), and it was analyzed using FlowJo version 10.6.1 (Miltenyi Biotec).

### Assessing oxygen-dependent Hb polymer content in single RBCs
Hb polymer content was assessed in individual RBCs using a previously reported quantitative absorptive cytometer (QAC) system[35]. The QAC system images individual RBCs and infers Hb polymer content based on the principle that the presence of Hb polymer lowers RBC oxygen saturation in proportion to polymer concentration[40, 41]. At a fixed oxygen tension, an RBC's hemoglobin−oxygen saturation will decrease if some of its hemoglobin polymerizes. Among a population of RBCs at the same fixed oxygen tension, those with significant amounts of polymer will have significantly lower oxygen saturation. RBCs with significant amounts of polymer will also typically become distorted and "sickled". The QAC utilizes both single-RBC oxygen saturation and single-RBC morphology to detect the presence of Hb polymer in single RBCs.

### Assessing F-cell levels and Hb fractions
We assessed F-cell levels using flow cytometry (BDFACS Lyric) with a previously published protocol[37]. RBCs were fixed, permeabilized, and stained with anti-HbF antibody (MHFH04, R-PE, life technologies) at a final concentration of 12.5 μg/mL. Post incubation and washes, the samples were acquired using the BD FACSLyric RUO instrument and BD FACS FACSuite RUO v.1.5 (Beckton Dickinson). The F-cell reports

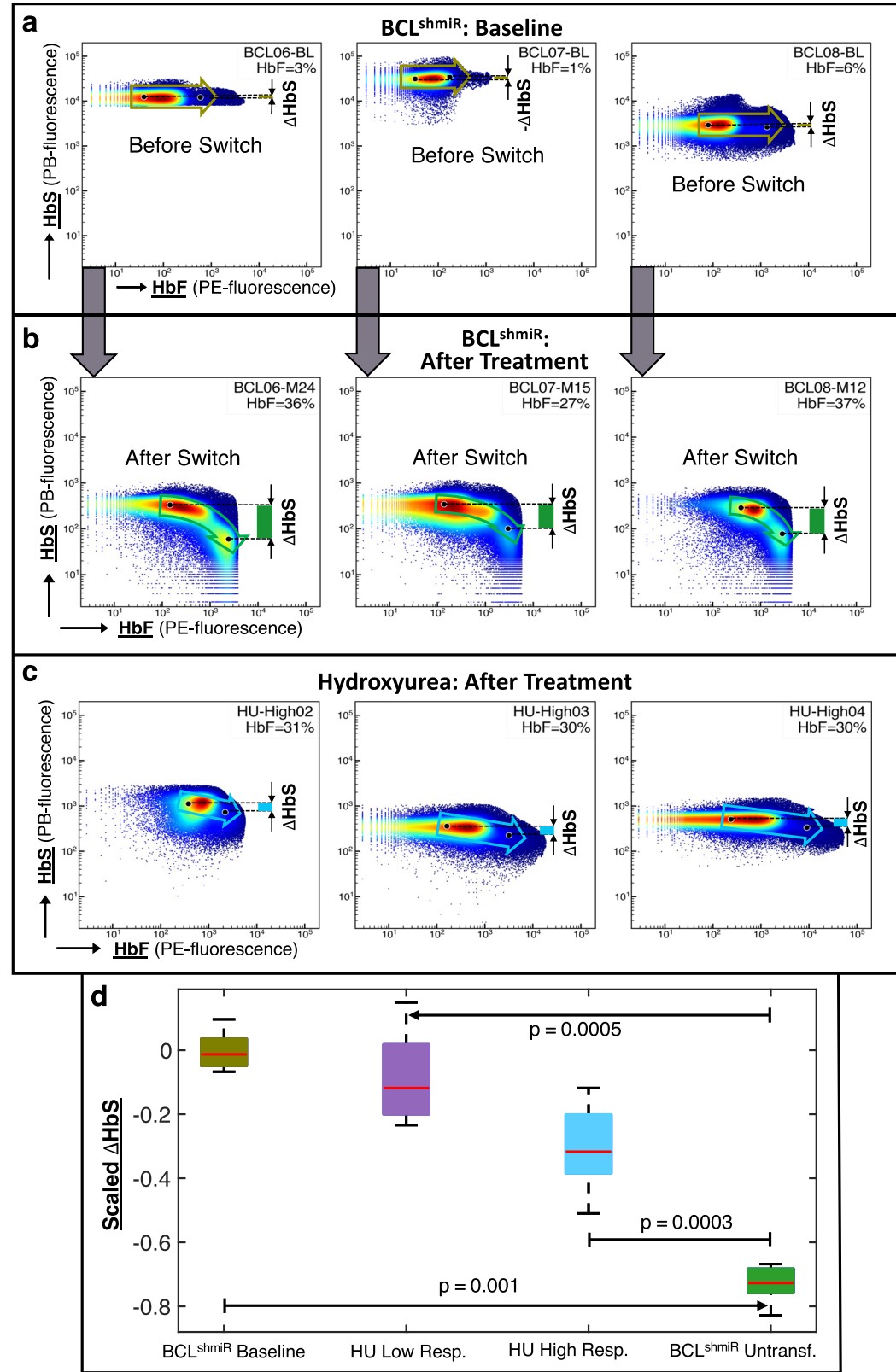

were generated using the BD FACSuite software. The positive cutoff point was set at 0.1% above the negative population of unstained control RBCs. Fractions of whole blood HbF and HbS with respect to the total Hb were measured in each blood sample by high-performance liquid chromatography (HPLC). The gating strategy applied to exclude doublets and to determine the F cell fraction is described in Supplementary Fig. S7.

**Data analysis, statistics, and reproducibility**

Box plots compare the results between patients treated with gene therapy and HU. Boxes show median (central mark), 25th percentile (bottom edge), and 75th percentile (top edge). Top whiskers extend to the highest data point that is below or equal to the 75th percentile plus 1.5 times the IQR, while the bottom whiskers extend to the lowest data point that is above or equal to the 25th percentile minus 1.5 times the

**Fig. 4 | Hb switching after BCL11A inhibition leads to high HbF/low HbS-containing cells.** We investigated the mechanism of Hb switching in BCL[shmiR] and HU-treated patients by performing intracellular co-staining of single RBCs to assess their relative levels of HbS and HbF. In panels **a**–**c** the flow cytometry representative dot plots show the HbS fluorescence intensity (y-axis) as a function of HbF fluorescence intensity (x-axis) after intracellular staining for BCL[shmiR] at baseline (**a**) and ≥12 months after drug product infusion (**b**). Panel **c** shows representative measurements performed on HU-treated patients. Flow cytometry data was collected for BCL[shmiR]-patients at baseline (n = 5), Untransfused BCL[shmiR] ≥12 months after drug product infusion (n = 7), HU High Responders (n = 8), and HU Low Responders (n = 6). Two BCL[shmiR] baseline blood samples and one HU High Responder blood sample were not available for this assay. For each data sample available, the cells were ordered by HbS fluorescence intensity and clustered into quartiles. The average HbS and HbF fluorescence intensities of each quartile were computed and scaled by dividing the average intensities by their maximum quartile values. Left and right black dots in panels **a** to **c** respectively indicate the average of scaled HbF and HbS fluorescence intensities from the 1st and 4th quartiles, while the arrow indicates the Hb switch from the 1st to 4th quartile, and green bars indicate variation in HbS intensities between 4th and 1st quartile. Panel **d** shows boxplots comparing scaled HbS fluorescence intensity between its 4th and 1st quartile for all four cohorts. More than 100,000 cells were collected for each blood sample. Boxplot properties and the method used to compute the p-values in this study are described in the "Methods" subsection "Data analysis, statistics, and reproducibility". Source data are provided as a Source Data file.

IQR. Black diamonds represent outliers. The p-values shown in the box plots were computed by applying the Wilcoxon rank sum one-sided test, which is equivalent to the Mann–Whitney U one-sided test, with the subroutine *ranksum* from MATLAB[42]. All measurements used in this study were taken from different blood samples. Flow cytometry data was analyzed using GraphPad PRISM (version 6.07) and Python (version 3.7.9).

## Regression model of HbF kinetics
Mixed effects linear regression modeling was used to estimate the evolution of the proportion of RBCs containing a minimum of 10 pg of HbF per cell in BCL[shmiR] as a function of time over 24 months after drug product infusion. The fitting was performed by using the MATLAB subroutine *fitlmematrix*[42].

## Reporting summary
Further information on research design is available in the Nature Portfolio Reporting Summary linked to this article.

## Data availability
All source data are provided as a Source Data File. The VCN values from patients 02–08 shown in Table S2 are reused from Esrick et al.[22]. All relevant trial documentation is provided in "Participants and treatment". This study only shares deidentified patient data. Source data are provided with this paper.

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

## Acknowledgements

We thank Carlo Brugnara, David G. Nathan, Franklin Bunn, Matthew Heeney, and Orah Platt for helpful discussions and for critical review of an earlier version of the manuscript, Martin Steinberg and Swee Lay Thein for helpful discussions, France Pirenne (EFS) for her advice, and Marie Cambot (INNOVHEM) for providing the best-controlled batches of antibodies and data about reproducibility and repeatability of the HbF/cell method. This work was supported by the National Institutes of Health (R01HL 137848, D.A.W.; R01HL 132906, D.K.W.; R01 HL 158102, J.M.H.), the National Heart, Lung, and Blood Institute (Cure Sickle OT2HL154815, D.A.W.), by INSERM and the French Blood Establishment (EFS), and portions of this work were conducted in the Minnesota Nano Center, which is supported by the National Science Foundation through the National Nanotechnology Coordinated Infrastructure (NNCI) under Award Number ECCS-2025124 (D.K.W.).

## Author contributions

D.C.D.S., N.H., E.B.E., M.F.C., N.M.A., M.A. E.A., C.B., G.D.C., F.G., D.L., A.M., E.M., E.S., D.W., D.K.W., D.A.W., P.B., and J.M.H. contributed to this study, planning analysis, collecting data, discussing results, and participated in drafting or revising the manuscript.

## Competing interests

The following authors declare the following relationships which may be considered as potential competing interests: E.B.E.: bluebird bio: Consultancy. D.A.W.: Emerging Therapy Solutions: Chief Scientific Chair (consulting position, ended in 2021); Skyline Therapeutics (formerly Geneception): Scientific Advisory Board; BioMarin: Insertion Site Advisory Board; Novartis: Committee, Novartis ETB115E2201 (eltrombopag in aplastic anemia). Advisory fees donated to NAPAAC; Alerion Biosciences: Co-founder (now licensed to Avro Bio, potential for future milestones/royalties); Beam Therapeutics: Scientific Advisory Board; Orchard Therapeutics: Scientific Advisory Board (position ended 05/20/2021) and co-founder. bluebird bio provided GMP vector for a clinical trial (sickle cell disease) and Insertion Site Analysis Advisory Board. P.B.: Consultant for ADDMEDICA, Novartis, ROCHE, GBT, bluebird bio, EMMAUS, HEMANEXT, AGIOS, VERTEX; Lecture fees for Novartis, ADDMEDICA, AGIOS, JAZZPHARMA, VERTEX; Steering committee for Novartis and ADDMEDICA; Research support from ADDMEDICA, foundation Fabre, Novartis, bluebird bio, EMMAUS, GBT; Cofounder of INNOVHEM. The authors not listed here declare no competing interests.

## Additional information

Daniel C. De Souza[1,2,3,15], Nicolas Hebert[4,5,6,15], Erica B. Esrick[7,8,9], M. Felicia Ciuculescu[8], Natasha M. Archer[7,8,9], Myriam Armant [8], Étienne Audureau[10,11], Christian Brendel [7,8,9], Giuseppe Di Caprio [1,9,12,13], Frédéric Galactéros[5,6], Donghui Liu[8], Amanda McCabe[8], Emily Morris[8], Ethan Schonbrun[1], Dillon Williams[14], David K. Wood [14], David A. Williams[7,8,9,16] ✉, Pablo Bartolucci[5,6,16] ✉ & John M. Higgins [1,2,3,16] ✉

[1]Center for Systems Biology, Massachusetts General Hospital, Boston, MA, USA. [2]Department of Systems Biology, Harvard Medical School, Boston, MA, USA. [3]Department of Pathology, Massachusetts General Hospital, Boston, MA, USA. [4]French Blood Establishment (EFS), Créteil, France. [5]University Paris-Est-Créteil, IMRB, Laboratory of excellence LABEX, Créteil, France. [6]Paris-East Créteil University, Henri Mondor University Hospitals, APHP, Sickle Cell Referral Center—UMGGR, Créteil, France. [7]Dana-Farber/Boston Children's Cancer and Blood Disorders Center, Boston, MA, USA. [8]Boston Children's Hospital, Harvard Medical School, Boston, MA, USA. [9]Department of Pediatrics, Harvard Medical School, Boston, MA, USA. [10]INSERM U955 Team CEpiA, Paris-East Créteil University, Créteil, France. [11]Department of Public Health, Henri Mondor University Hospitals, APHP, Créteil, France. [12]Program in Cellular and Molecular Medicine, Boston Children's Hospital, Boston, MA, USA. [13]Department of Biomedical Engineering, University of Strathclyde, Glasgow, United Kingdom. [14]Department of Biomedical Engineering, University of Minnesota, Minneapolis, MN, USA. [15]These authors contributed equally: Daniel C. De Souza, Nicolas Hebert. [16]These authors jointly supervised this work: David A. Williams, Pablo Bartolucci, John M. Higgins. ✉e-mail: david.williams2@childrens.harvard.edu; pablo.bartolucci@aphp.fr; higgins.john@mgh.harvard.edu

