## [Peer Review file · Nature Communications]

REVIEWER COMMENTS

Reviewer #1 (Remarks to the Author):

This manuscript describes work to analyze the effects of therapeutic HbF induction for sickle cell disease at a single cell level. Specifically, the authors use newly developed flow cytometry and microfluidic techniques to measure the levels of HbF, HbS and polymerized HbS in single cells. They compare RBCs in patients treated successfully with hydroxyurea therapy to those treated with gene therapy to raise HbF by suppressing the expression of BCL11A. Using standard assays, investigators showed that both therapies resulted in similar %HbF levels in RBC lysates and the F-cell fraction was higher in HU responders. However, gene therapy produced higher %HbF per F cell with stronger reductions in HbS and greater reduction in HbS polymerization at low oxygen tension. These findings show how newly developed techniques for single cell analysis may be superior to current standards for predicting the clinical efficacy of therapeutic HbF induction. Ultimately, the question is whether these new biomarkers predict clinical outcome. In this regard, providing some clinical outcome data, such as frequency of pain crisis if possible, would strengthen the manuscript.

Other comments and suggestions:

1. A supplemental table should show hematological indices for each individual patient, including the HU treated ones (which is not currently shown). The table should also include WBC levels, if available. The table should also include for each patient the results of single cell studies shown in Figures 1-4.
2. In addition to showing summaries of cells that surpass estimated clinical thresholds, it would be useful to show graphs of the distribution of HbS and HbF concentrations across the entire RBC populations and within the population of F-cells.
3. Figure 1 describes HbF levels in single cells determined by a recently published flow cytometry method whereas Figure 3a shows a calculated average, correct? This should be explained better in the text. How well do the two methods correlate? Does single cell HbF quantification reveal an average HbF per cell concentration similar to what is reported in Figure 3a?

Related to above, the innovative techniques should be explained in a bit more detail so that the methods may be understood by a wide audience. Admittedly, these methods have been described elsewhere, but readers would benefit from a brief summary of the methods and their limitations. For example, the “polymerization assay” does not quantify HbS polymers, but rather, measures RBC shape

changes that result from polymerization. How are single cell HbS and HbF determinations performed? What is the accuracy and dynamic range of these methods?

4. The data indicate that HU causes induction of HbF without a reciprocal decrease in HbS expression. Does this suggest that HU effects are post transcriptional and is there evidence to support that? Is this consistent with RNA seq analysis of patient reticulocytes, either from this study or from published studies?

Minor

1. There are numerous typographical errors in the manuscript. Some sections are not easy to read. See major comment 3.

2. Page 7: for clarification, please explain the approach used to determine “the fraction of RBCs with > 10pg HbF” and the rationale for that threshold (which is currently explained later, on page 16).

3. Figure 2- it would be useful to show fraction of RBCs with no polymers at O₂ tension equal to or greater than 3.7%..

4. Figure S2c is referred to in the text but panel c is not indicated or discussed in the Figure S2 legend.

Reviewer #2 (Remarks to the Author):

In this study, the Authors analyze hemoglobin expression in HU-treated SCD patients and in SCD patients treated with hematopoietic stem cells (HSC) transduced with a lentivector downregulating BCL11A (a transcription factor downregulating fetal hemoglobin (HbF) expression). Although interesting, results seem incremental compared to previously reported studies. In particular:

- Most of the results and analyses shown in this study are “expected” based on the results reported in Table 1 (similar HbF expression with higher frequency of HbF cells in the high responder group)
- VCN as well as clinical findings (frequency of VOC, etc etc) should be reported in gene therapy-treated patients as well as clinical data in HU-treated patients

- Genotype (e.g., Alpha thalassemic trait) should be reported for each patient as well as all the different parameters analyzed in these patients

- Statements on hemoglobin switching in the discussion should be toned down, the Authors implicitly suggest that HU does not induce hemoglobin switching. Without showing any data at RNA levels, Authors cannot suggest that.

Reviewer #3 (Remarks to the Author):

The manuscript by De Souza, Hebert et al. compares the hematological effects 2 types of treatment of sickle cell disease. One is a gene therapy (GT) approach targeting BCL11A by RNAi to relieve the fetal/adult hemoglobin switch and the other is hydroxyurea (HU), the current standard of care also known to act by increasing fetal hemoglobin. The manuscript presents a detailed analysis of Hb content in red blood cells at the single cell level to precisely quantify HbF, HbS and total Hb as well as RBC sickling in response to deoxygenation. Their results show that GT effectively reversed the fetal to adult Hb switch. The 7/9 transfusion-independent patients treated by GT showed lower % of RBC with fetal Hb (F-cells) than patients treated by HU but their RBCs have higher % of HbF, with a lower % of HbS and greater resistance to deoxygenation-induced polymerization than in patients that are highly-responsive to HU. While GT concomitantly increased HbF and decreased HbS, HU did not decrease HbS content and therefore probably HU acts through a different mechanism than controlling the fetal/adult Hb switch.

Overall, this work is focused on hematology but it has a broader impact considering the public health and global health impact of sickle cell disease, and considering the multiple gene therapy approaches currently developed to treat this disease. This work is also of high importance due to the novelty of the technological analysis employed and because of the results obtained. However several points require attention.

Major comments :

The patient populations that are compared (GT vs HU) are not age-matched and the authors should show that this would not have an impact on the Hb level results. In Table 1 it would be useful to have data on a population of age-matched untreated controls, if possible.

For a more complete understanding, we are missing data comparing the pre- and post-GT results (some patients could be analysed as shown in Figure 4). Authors should specify if GT patients were also receiving HU prior to GT.

In Table 1, the layout is not completely clear and the first column should mention the cohort in which the patients belong.

Also, Table 1 contains at the same time individual GT patient data and averages on the HU population. It would be useful to show individual data on all patients analyzed (GT and HU) and perhaps a second Table with only averages and statistics. The reviewer realizes that the individual data Table would be quite large, so it could be perhaps shown in supplemental information, but this would help to define the different groups (transfusion dependent, high or low responder).

Could the authors speculate on why there are low HU responders ?

A novel finding of this study is to show the differential effects of HU and GT on HbS levels. Could the authors expand the discussion and try to briefly speculate on possible mechanisms of action of HU ?

The % F-cells is lower in GT than HU because it is presumably correlated to the level of engraftment of gene-corrected stem cells. Is this correct ? Could this point be discussed because it would be misleading to conclude that GT works better than HU. In GT low engraftment of gene-corrected stem cells can occur. Is this the case of the 2 out of 9 patients who remain transfusion dependent ? *

Minor : Text on page 5 indicates F/U of 17-49 months and Table 1 shows 16-48. Please harmonize or include the most recent data in Table 1.

REVIEWER COMMENTS

Reviewer #1 (Remarks to the Author):

This manuscript describes work to analyze the effects of therapeutic HbF induction for sickle cell disease at a single cell level. Specifically, the authors use newly developed flow cytometry and microfluidic techniques to measure the levels of HbF, HbS and polymerized HbS in single cells. They compare RBCs in patients treated successfully with hydroxyurea therapy to those treated with gene therapy to raise HbF by suppressing the expression of BCL11A. Using standard assays, investigators showed that both therapies resulted in similar %HbF levels in RBC lysates and the F-cell fraction was higher in HU responders. However, gene therapy produced higher %HbF per F cell with stronger reductions in HbS and greater reduction in HbS polymerization at low oxygen tension. These findings show how newly developed techniques for single cell analysis may be superior to current standards for predicting the clinical efficacy of therapeutic HbF induction. Ultimately, the question is whether these new biomarkers predict clinical outcome. In this regard, providing some clinical outcome data, such as frequency of pain crisis if possible, would strengthen the manuscript.

We are pleased the reviewer finds that our single-cell assays and analysis have identified treatment differences at the single-cell level. We agree with the Reviewer that these novel findings provide strong rationale for a larger follow-up study designed to determine whether these single-cell characteristics predict clinical outcomes better than current standards.

The current study was not designed to answer that question, and study subjects were selected solely on the basis of HbF% without regard for clinical outcomes. Complete clinical histories and detailed clinical outcome data are not available for all subjects. Review of available records find that the 12 HU High patients were admitted or seen in an emergency department for IV analgesia a median of 0 times (range 0-2) during the median of 2 years (range 1-5) of hydroxyurea use while under care at our study hospital. Because unbiased statistical comparison of any clinical characteristics of the patient groups and extraction of accurate pre- and post-hydroxyurea clinical data for the High HU group is not possible, we are not including this clinical data in the manuscript.

Other comments and suggestions:

1. A supplemental table should show hematological indices for each individual patient, including the HU treated ones (which is not currently shown). The table should also include WBC levels, if available. The table should also include for each patient the results of single cell studies shown in Figures 1-4.

We have added all of this individual patient data to our revised manuscript in Tables 1 and S1.

2. In addition to showing summaries of cells that surpass estimated clinical thresholds, it would be useful to show graphs of the distribution of HbS and HbF concentrations across the entire RBC populations and within the population of F-cells.

Both the single-cell method that utilizes intracellular staining of HbF to measure the fraction of RBCs surpassing HbF mass thresholds, and the method that utilizes intracellular co-staining of HbS and HbF to assess relative levels of HbS in RBCs do not provide simultaneous measurements of single-RBC volume nor of single-RBC HbS mass. Single-RBC HbS and HbF concentrations are therefore not currently available, but we strongly agree with the reviewer that the results in our paper highlight the potential insight those measurements would provide, and the development of well-validated assays to measure these single-cell concentrations is an important direction for the field.

Our method provides accurate and reproducible quantification of HbF mass per RBC. However it does not currently measure the mass of HbS per RBC as reliably as it does for HbF. Similarly, it is extremely difficult to measure the volume per cell accurately by flow cytometry or imaging flow cytometry, especially for RBCs from sickle cell patients. While concentration of HbF or HbS per cell is not available, we have shown that there is strong correlation between the mean cellular hemoglobin mass (MCH) and volume (MCV), with the regression line corresponding to the MCHC (Higgins and Mahadevan, PNAS 2010). This correlation is replicated in sickle cell patients as shown in the top panel below (n=999 SS patients, Bartolucci personal data) and with data from the BCL^{shmiR}, HU High Responder, and HU Low Responder cohorts in the bottom panel.

3. Figure 1 describes HbF levels in single cells determined by a recently published flow cytometry method whereas Figure 3a shows a calculated average, correct? This should be explained better in the text. How well do the two methods correlate? Does single cell HbF quantification reveal an average HbF per cell concentration similar to what is reported in Figure 3a?

We appreciate the suggestion and have clarified the differences between those assays in our revised Results section. Briefly, Fig. 1a shows results collected with recently-developed flow cytometry assay that is calibrated for measurement of single-RBC HbF mass, while Fig. 3a shows results collected with a standard HPLC assay for total HbF% and a standard FACS assay for F-cell%. The F-cell threshold of the assay used in our study has been estimated to be 4.8 pg +/- 2.2. The figure below shows strong correlation between the F-cell level measured by this assay and the percentage of RBCs with > 4pg HbF as determined by the single-RBC HbF mass assay.

Related to above, the innovative techniques should be explained in a bit more detail so that the methods may be understood by a wide audience. Admittedly, these methods have been described elsewhere, but readers would benefit from a brief summary of the methods and their limitations. For example, the “polymerization assay” does not quantify HbS polymers, but rather, measures RBC shape changes that result from polymerization. How are single cell HbS and HbF determinations performed? What is the accuracy and dynamic range of these methods?

We are pleased that the reviewer finds our techniques innovative, and we have expanded our description of these assays in our revised Methods and Results sections.

4. The data indicate that HU causes induction of HbF without a reciprocal decrease in HbS expression. Does this suggest that HU effects are post transcriptional and is there evidence to support that? Is this consistent with RNA seq analysis of patient reticulocytes, either from this study or from published studies?

The reviewer raises important questions about the molecular mechanism for the induction of HbF by HU. We have added a citation to Macklis et al. and Dover et al. showing differential expression of HbF in BFUe and CFUe derived red cells with the implication that HU works by selectively killing CFUe forcing more immature BFUe to contribute to circulating red cells. We are not aware of any follow-up studies that confirmed this differential sensitivity or otherwise shows definitively the mechanism of HU leading to elevated HbF, but our current hypothesis is that this explains HU effects. As the reviewer notes, RNA seq analysis of reticulocytes would be useful to determine whether HU effects are post-transcriptional. We are not aware of any published RNA seq data that would help answer this question. Single-cell RNA sequence analysis is not within the scope of the current study but is a high priority for future investigation.

Minor

1. There are numerous typographical errors in the manuscript. Some sections are not easy to read. See major comment 3.

We appreciate the careful reading and have fixed several typographical errors and have revised the text referenced in major comment 3. None of our conclusions has changed, but we hope that readability has been enhanced.

2. Page 7: for clarification, please explain the approach used to determine “the fraction of RBCs with > 10pg HbF” and the rationale for that threshold (which is currently explained later, on page 16).

We have expanded our description of the single-RBC HbF mass measurement methodology in the revised Methods section and following the reviewer’s suggestion have moved our initial description of this method and the rationale for the selected threshold earlier in the paper. As noted in the revised paper, the rationale for this threshold is present in prior publications by Martin Steinberg and others, which we cite.

3. Figure 2- it would be useful to show fraction of RBCs with no polymers at O₂ tension equal to or greater than 3.7%.

This result was noted in the Results section of our original submission, and we agree with the reviewer that is helpful to show that data in a figure, which is included below and in the Supplementary Information of our revised manuscript in Figure S2.

Figure S1 **Fractions of RBCs with no detectable Hb polymer content after BCL11A inhibition or HU treatment.** The percentage of RBCs that have no detectable Hb polymer content at 3.7% oxygen tension was measured in vitro in Untransfused BCL^{shmiR} (n = 7; green boxes), in HU High Responders (n = 10; blue boxes), and in HU Low Responders (n = 8; purple boxes).

Reviewer #2 (Remarks to the Author):

In this study, the Authors analyze hemoglobin expression in HU-treated SCD patients and in SCD patients treated with hematopoietic stem cells (HSC) transduced with a lentivector downregulating BCL11A (a transcription factor downregulating fetal hemoglobin (HbF) expression). Although interesting, results seem incremental compared to previously reported studies. In particular:

- Most of the results and analyses shown in this study are “expected” based on the results reported in Table 1 (similar HbF expression with higher frequency of HbF cells in the high responder group)

We are glad the reviewer finds our results “interesting.” We note that our study is the first to analyze the single-cell effects of a gene therapy that reverses the fetal to adult hemoglobin switch in sickle cell patients. We are pleased that our novel single-cell techniques and analysis are consistent with the Reviewer’s expectation, and we agree with the assessment of Reviewers #1 and #3 that our results are of “high importance due to the novelty of the technological analysis employed” and “may be superior to current standards for predicting the clinical efficacy of therapeutic HbF induction.”

- VCN as well as clinical findings (frequency of VOC, etc etc) should be reported in gene therapy-treated patients as well as clinical data in HU-treated patients

VCN data is now provided in Table S2 as requested. We agree with this Reviewer and the others that our study’s identification of single-cell differences between treatments provides strong rationale for a larger follow-up study designed to determine whether these differences in single-cell characteristics predict clinical outcomes better than current standards. As detailed above in our response to Reviewer #1, the current study was not designed to answer that question. Study subjects were selected solely on the basis of HbF% without regard for clinical outcomes, and complete clinical histories and detailed clinical outcome data are not available for all subjects. Unbiased statistical comparison of any clinical characteristics of the patient groups and extraction of accurate pre- and post-hydroxyurea clinical data for the High HU group is not possible.

- Genotype (e.g., Alpha thalassemic trait) should be reported for each patient as well as all the different parameters analyzed in these patients

We have added detailed laboratory parameters for all subjects in all three cohorts in our revised manuscript in Tables 1 and S1. This study matched patients solely on the basis of HbF%, and no genotype information was used, including alpha globin. Alpha globin genotype information is not available for some HU-treated patients. Investigating a potential influence of alpha thalassemia status is a priority in future studies and is included in the ongoing NIH-funded phase 2 GRASP trial (NCT 05353647).

- Statements on hemoglobin switching in the discussion should be toned down, the Authors implicitly suggest that HU does not induce hemoglobin switching. Without showing any data at RNA levels, Authors cannot suggest that.

We agree with the Reviewer about the importance of molecular data for elucidating HU’s mechanism. Our initial submission stated that the mechanism of action of HU was undefined, and we have expanded our Discussion based on the Reviewer’s suggestion to cite prior studies showing differential expression of HbF in BFUe and CFUe derived red cells with the implication that HU works by selectively killing CFUe forcing more immature BFUe to contribute to circulating red cells. We are not aware of any follow-up studies that confirmed this differential sensitivity or otherwise shows definitively the mechanism of HU leading to elevated HbF, but our current hypothesis is that this explains HU effects. We make clear that RNA or other molecular data would be required to determine to what extent HU induces any hemoglobin switching.

Reviewer #3 (Remarks to the Author):

The manuscript by De Souza, Hebert et al. compares the hematological effects 2 types of treatment of sickle cell disease. One is a gene therapy (GT) approach targeting BCL11A by RNAI to relieve the fetal/adult hemoglobin switch and the other is hydroxyurea (HU), the current standard of care also known to act by increasing fetal hemoglobin. The manuscript presents a detailed analysis of Hb content in red blood cells at the single cell level to precisely quantify HbF, HbS and total Hb as well as RBC sickling in response to deoxygenation. Their results show that GT effectively reversed the fetal to adult Hb switch. The 7/9 transfusion-independent patients treated by GT showed lower % of RBC with fetal Hb (F-cells) than patients treated by HU but their RBCs have higher % of HbF, with a lower % of HbS and greater resistance to deoxygenation-induced polymerization than in patients that are highly-responsive to HU. While GT concomitantly increased HbF and decreased HbS, HU did not decrease HbS content and therefore probably HU acts through a different mechanism than controlling the fetal/adult Hb switch.

Overall, this work is focused on hematology but it has a broader impact considering the public health and global health impact of sickle cell disease, and considering the multiple gene therapy approaches currently developed to treat this disease. This work is also of high importance due to the novelty of the technological analysis employed and because of the results obtained. However several points require attention.

We are pleased the Reviewer feels our work is of high importance due to both novelty of analysis and the nature of the results.

Major comments :

The patient populations that are compared (GT vs HU) are not age-matched and the authors should show that this would not have an impact on the Hb level results. In Table 1 it would be useful to have data on a population of age-matched untreated controls, if possible.

The Reviewer is correct that patient populations were selected and matched solely on the basis of HbF%, and other factors including patient age were not considered. We have reviewed the literature for well-designed studies reporting age-stratified and treatment-stratified reference intervals for Hb among individuals with sickle cell disease and have noted in our revised manuscript that our results are consistent, for instance with Kinney et al. 1999 which reported a similar range of Hb (median 7.8 g/dL) in 84 untreated and untransfused sickle cell patients with a median age of 9.1 years old, but studies designed to define these reference intervals are limited, and age-stratified and treatment-stratified reference intervals for Hb among individuals with sickle cell disease are not well-defined in general.

For a more complete understanding, we are missing data comparing the pre- and post-GT results (some patients could be analysed as shown in Figure 4). Authors should specify if GT patients were also receiving HU prior to GT.

We have added information on HU status of GT subjects prior to GT in our revised manuscript.

In Table 1, the layout is not completely clear and the first column should mention the cohort in which the patients belong.

We appreciate the suggestion and have added the helpful column.

Also, Table 1 contains at the same time individual GT patient data and averages on the HU population. It would be useful to show individual data on all patients analyzed (GT and HU) and perhaps a second Table with only averages and statistics. The reviewer realizes that the individual data Table would be quite large, so it could be perhaps shown in supplemental information, but this would help to define the different groups (transfusion dependent, high or low responder).

We have added individual data for all HU-treated patients and all BCL_{shmiR} patients for all Figures in Table S1.

Could the authors speculate on why there are low HU responders ?

The reviewer asks an important question that has challenged the field. As noted above, the molecular mechanism for HU is not fully defined, and that gap in understanding adds to confusion. Genetic variants may be involved, and recent studies exploring pharmacokinetics-guided dose escalation suggest that heterogeneity in pharmacokinetics may be another contributor.

A novel finding of this study is to show the differential effects of HU and GT on HbS levels. Could the authors expand the discussion and try to briefly speculate on possible mechanisms of action of HU ?

We are grateful for reviewer's suggestion. We agree that our results showing significant differences in treatment effect for HU and BCL11A knockdown at the single cell level will motivate and guide future studies of the molecular mechanisms of hydroxyurea. Without the benefit of molecular data, and in keeping with suggestions of other reviewers, we have revised our discussion to emphasize that the mechanism of action of HU is undefined. While our data is consistent with the hypothesis that HU does not significantly reverse the fetal to adult switch, further studies with RNA sequence analysis and other molecular data are required to verify this hypothesis.

The % F-cells is lower in GT than HU because it is presumably correlated to the level of engraftment of gene-corrected stem cells. Is this correct ? Could this point be discussed because it would be misleading to conclude that GT works better than HU. In GT low engraftment of gene-corrected stem cells can occur. Is this the case of the 2 out of 9 patients who remain transfusion dependent ? *

We agree with the Reviewer's hypothesis that level of engraftment is likely related to %F-cells in GT subjects. Another factor is the efficiency with which the genetic alteration itself leads to HbF induction. It should also be noted, as is detailed in Esrick et al., that one of the GT patients was maintained on a pre-determined transfusion protocol that had been agreed upon prior to study

enrollment based on clinical history of moyamoya. Detailed molecular studies would be needed for definitive answers to the Reviewer's questions and are a priority for future investigation.

Minor : Text on page 5 indicates F/U of 17-49 months and Table 1 shows 16-48. Please harmonize or include the most recent data in Table 1.

We appreciate the careful reading and have corrected those statements in our revised submission.

REVIEWERS' COMMENTS

Reviewer #1 (Remarks to the Author):

I am satisfied with the authors' responses to all of my questions except one. Regarding the response to question 2 (Reviewer 1) requesting that Hb expression be displayed as continuous variables across the RBC populations: The investigators perform single cell methods to assess HbF expression. It is not clear to me why HbF determinations (particularly mass) are shown as categorical thresholds and cannot be compared as a continuous variable on a per cell basis. If true, this represents a weakness of this paper that should be explained in the text. A distribution is more useful than a threshold because the true threshold level of HbF per cell for prevention of sickling is not known.

Reviewer #2 (Remarks to the Author):

I thank the Authors for providing VCN data at month 6, the Authors should state if these values were stable over time (including at the timepoints when hemoglobin expression was analyzed). I strongly suggest the Authors to report clinical data and genotype at least for the gene therapy treated patients and the patients treated with HU for whom data can be retrieved. Without clinical data, some parts of the the study remains inconclusive and, in some cases, potentially misleading.

Reviewer #3 (Remarks to the Author):

The revised manuscript has been improved and the authors answered most of my questions. However, I still have two comments which should be addressed prior to publication.

1° I still have an issue with Table 1. It provides descriptive statistics for the HU groups whereas it shows individual patient data for the gene therapy group, which is illogical and suggests that the hematological parameters of the gene therapy patients are more valuable than those of the HU group. Since all the individual patients data are reported in Table S1, the gene therapy patient data are duplicated unnecessarily. My suggestion is to present in Table 1 all the groups through descriptive statistics and to retain the detailed individual patients data in Table S1. The tables should also indicate how many data points are used to generate median values. The legend says that the data are median values for data points greater than 5 months but I don't understand what this means. Do you mean data obtained after at least 5 months of treatment ? How many data points ?

2° Some of the methods used are poorly described. It remains difficult to know if the measures of scaled delta HbF/HbS values could be replicated elsewhere (is the term "scaled" a good choice of words?). From what is explained in the legend to Figure 4 it comes from computed normalized values of FACS signal but which software or script is used to calculate them ? This should be explained in the Materials and methods section rather than figure legend.

Subject: Final revisions for Nature Communications manuscript NCOMMS-22-51506A

REVIEWERS' COMMENTS

Reviewer #1 (Remarks to the Author):

I am satisfied with the authors' responses to all of my questions except one. Regarding the response to question 2 (Reviewer 1) requesting that Hb expression be displayed as continuous variables across the RBC populations: The investigators perform single cell methods to assess HbF expression. It is not clear to me why HbF determinations (particularly mass) are shown as categorical thresholds and cannot be compared as a continuous variable on a per cell basis. If true, this represents a weakness of this paper that should be explained in the text. A distribution is more useful than a threshold because the true threshold level of HbF per cell for prevention of sickling is not known.

We are glad that Reviewer is satisfied with our responses and appreciate the opportunity to provide further detail on the single-cell HbF determinations in the manuscript. The paper shows results relative to 10pg thresholds because that threshold was proposed by Steinberg and others (Maier-Redelsperger et al., 1994, Ngo et al., 2012, Steinberg et al., 2014). We strongly agree with the reviewer that thresholds for prevention of sickling are not well-defined and will depend on multiple factors including HbF per cell (Figure 1) and also importantly on oxygen independent of HbF (Figure 2). Our focus in this manuscript is to study the cellular level differences between these two treatments and investigate the potential relationship to globin switching. We agree with the reviewer that another important topic is understanding which levels of HbF prevent sickling at different oxygen tensions (and different MCHC and other factors), and that's an important direction for future studies.

Reviewer #2 (Remarks to the Author):

I thank the Authors for providing VCN data at month 6, the Authors should state if these values were stable over time (including at the timepoints when hemoglobin expression was analyzed).

We thank the Reviewer and for the good suggestion and have adding a statement in our revised manuscript that the BCL^{shmiR} patients did demonstrate stable VCNs in both peripheral blood and bone marrow after 6 months as shown in Table S2.

I strongly suggest the Authors to report clinical data and genotype at least for the gene therapy treated patients and the patients treated with HU for whom data can be retrieved. Without clinical data, some parts of the the study remains inconclusive and, in some cases, potentially misleading.

We agree with the reviewer that a clinical study that was designed and powered to compare these single-cell analyses with clinical outcomes would be very interesting and is a high priority for future investigation. However, since the current study was not designed for this purpose,

including clinical outcome data only for the patients for whom it happened to available could be very misleading. Because no clinical outcome data is included in our manuscript, we are careful to make no claims regarding the superiority of these single-cell analyses for predicting clinical outcomes compared to current standards.

Reviewer #3 (Remarks to the Author):

The revised manuscript has been improved and the authors answered most of my questions. However, I still have two comments which should be addressed prior to publication.

1° I still have an issue with Table 1. It provides descriptive statistics for the HU groups whereas it shows individual patient data for the gene therapy group, which is illogical and suggests that the hematological parameters of the gene therapy patients are more valuable than those of the HU group. Since all the individual patients data are reported in Table S1, the gene therapy patient data are duplicated unnecessarily. My suggestion is to present in Table 1 all the groups through descriptive statistics and to retain the detailed individual patients data in Table S1. The tables should also indicate how many data points are used to generate median values. The legend says that the data are median values for data points greater than 5 months but I don't understand what this means. Do you mean data obtained after at least 5 months of treatment ? How many data points ?

We are pleased the Reviewer feels the manuscript has been improved. We revised Table 1 to show how many data points were involved in the median calculations. Because of the novelty of the gene therapy and the collection of data at multiple time points, readers will benefit from the seeing that patient-level data in the main text, while a summary of the single timepoint data for the HU-treated patients is sufficient given space constraints, and we include full detail in the supplement for reference if needed.

2° Some of the methods used are poorly described. It remains difficult to know if the measures of scaled delta HbF/HbS values could be replicated elsewhere (is the term "scaled" a good choice of words?). From what is explained in the legend to Figure 4 it comes from computed normalized values of FACS signal but which software or script is used to calculate them ? This should be explained in the Materials and methods section rather than figure legend.

We have expanded our description of these methods as suggested in the Figure 4 caption and the Methods section of our revised manuscript, and further detail is provided in the reporting summary that will accompany the manuscript.